

# Sub-chronic testosterone treatment increases the levels of epithelial sodium channel (ENaC)-α, β and γ in the kidney of orchidectomized adult male Sprague–Dawley rats

Su Yi Loh,  Nelli Giribabu and  Naguib Salleh

Department of Physiology, University of Malaya, Kuala Lumpur, Malaysia

## ABSTRACT

Testosterone has been reported to cause blood pressure to increase. However mechanisms that underlie the effect of this hormone on this physiological parameter are currently not well understood. The aims of this study were to investigate effects of testosterone on expression of $\alpha, \beta$ and $\gamma$-epithelial sodium channel (ENaC) proteins and messenger RNAs (mRNAs) in kidneys, the channel known to be involved in Na$^+$ reabsorption, which subsequently can affect the blood pressure.

**Methods.** Adult male Sprague–Dawley (SD) rats were orchidectomized fourteen days prior to receiving seven days treatment with testosterone propionate (125 μg/kg/day or 250 μg/kg/day) with or without flutamide (androgen receptor blocker) or finasteride (5α-reductase inhibitor). Following sacrifice, the kidneys were removed and were subjected for $\alpha$, $\beta$ and $\gamma$-ENaC protein and mRNA expression analyses by Western blotting and Real-time PCR (qPCR) respectively. The distribution of $\alpha, \beta$ and $\gamma$-ENaC proteins in kidneys were observed by immunofluorescence.

**Results.** The $\alpha, \beta$ and $\gamma$-ENaC proteins and mRNA levels in kidneys were enhanced in rats which received testosterone-only treatment. In these rats, $\alpha, \beta$ and $\gamma$-ENaC proteins were distributed in the distal tubules and collecting ducts of the nephrons. Co-treatment with flutamide or finasteride resulted in the levels of $\alpha, \beta$ and $\gamma$-ENaC proteins and mRNAs in kidneys to decrease. In conclusions, increases in $\alpha, \beta$ and $\gamma$-ENaC protein and mRNA levels in kidneys mainly in the distal tubules and collecting ducts under testosterone influence might lead to enhance Na$^+$ reabsorption which subsequently might cause an increase in blood pressure.

## INTRODUCTION

Epithelial sodium channel (ENaC), which consists of three homologous subunits ($\alpha$, $\beta$ and $\gamma$) plays an important role in Na$^+$ reabsorption in kidneys (*Warnock & Rossier*, *2005*). The $\alpha$-subunit is required for Na$^+$ conductance while $\beta$- and $\gamma$-subunits are needed to enhance the channel activity (*Warnock & Rossier*, *2005*). Expression of ENaC in kidneys was under aldosterone influence (*Garty*, *2000*). Besides aldosterone, other hormones that could influence kidney ENaC expression include insulin and vasopressin (*Schild*, *2010*;

Corresponding author
Naguib Salleh,
naguib.salleh@yahoo.com.my

*Kamynina & Staub*, *2002*). Mutation of ENaC gene could lead to hypotension, while its prolonged activation could lead to severe hypertension (*Bubien*, *2010*). Expression of ENaC in kidneys could also be affected by sex hormones i.e., estrogen and progesterone (*Gambling et al.*, *2004*). There were evidences which suggest the involvement of testosterone in regulating kidney ENaC expression. Quan and colleagues (*2004*) reported that dihydrotestosterone (DHT) injection to adult male Sprague–Dawley rats could increase $Na^+$ reabsorption in kidney proximal tubules, suggesting that this could be mediated via ENaC. Meanwhile, administration of testosterone in spontaneous hypertensive (SHR) rats was found to decrease the pressure-induced natriuresis, again pointing towards the involvement of ENaC (*Reckelhoff, Zhang & Granger*, *1998*).

To date, the information with regard to effect of testosterone on ENaC expression in kidneys were far from complete. *Quinkler et al.* (*2005*) reported that sub-chronic (14 days) treatment of adult male Wistar rats with testosterone resulted in elevated *α-Enac* mRNA levels in kidney homogenates. Meanwhile, administration of dihydrotestosterone (DHT) causing the same effects, but was lesser than testosterone. Additionally, their study also showed that incubation of human kidney proximal tubule cell line with testosterone but not DHT caused increase in *α-ENaC* mRNA level with testosterone effect was being antagonized by flutamide. However, no expression for *β* and *γ*-ENaC was detected in these cells. In the meantime, testosterone and DHT were found to decrease expression of *α*, *β* and *γ*-ENaC subunits in ovariectomised female Wistar rat kidney (*Kienitz et al.*, *2009*). Based on these findings, we hypothesized that testosterone affects expression of all ENaC subunits i.e., *α*, *β* and *γ* in the kidneys, in which their co-existence will lead to a fully functioning ENaC channel, thus would augment $Na^+$ reabsorption which might result in the rise in blood pressure under testosterone influence (*Hanukoglu & Hanukoglu*, *2016*). Therefore, the aims of this study were to investigate testosterone effect on *α*, *β* and *γ* ENaC protein and mRNA expression levels in kidneys. In addition, the possible involvement of androgen receptor and DHT in mediating testosterone effects were also investigated.

## MATERIALS AND METHODS

### Animal preparation and hormonal treatment

Eight weeks old male Sprague–Dawley (SD) rats were housed in a clean and well ventilated environment under light-dark cycle (12/12 h). The animals were fed with standard rat diet (Harlan, Germany) and tap water *ad libitum*. All procedures were approved by Institutional Animal Care and Use Committee (IACUC), University of Malaya (ethics number: 2014-05-07/physio/R/NS). Sham operation and orchidectomies were performed under ketamine/xylazine anesthesia. Two (2) weeks following recovery, hormonal treatments were initiated, which coincide with the age of the rats at 10 weeks old. Rats were divided into eight (8) treatment groups with eight (8) rats per group and were given the following treatment for seven (7) consecutive days.

*Group 1*: Intact, sham-operated receiving peanut oil only—S

*Group 2*: Orchiectomized, receiving peanut oil only—O

*Group 3*: Orchiectomized, receiving 125 µg/kg/day testosterone—T125

*Group 4*: Orchiectomized, receiving 250 µg/kg/day testosterone—T250

*Group 5*: Orchiectomized, receiving 125 µg/kg/day testosterone plus flutamide (8 mg/kg/day)—T125+FU

*Group 6*: Orchiectomized, receiving 250 µg/kg/day testosterone plus flutamide (8 mg/kg/day)—T250+FU

*Group 7*: Orchiectomized, receiving 125 µg/kg/day testosterone plus finasteride (5 mg/kg/day)—T125+FN

*Group 8*: Orchiectomized, receiving 250 µg/kg/day testosterone plus finasteride (5 mg/kg/day)—T250+FN

At the end of the treatment, rats were sacrificed by cervical dislocation and kidneys were removed for molecular biological and histological analyses.

## Quantification of *Enac* mRNA levels in kidneys by high throughput qPCR (Fluidigm)

Immediately following removal, the kidneys from $n = 4$ rats per group were stored in RNAlater solution (Ambion, Foster City, CA, USA) in order to preserve the RNA integrity. Tissues were then weighted and disrupted in a tissue lysis buffer by using a rotor-stator homogenizer (Heidolph DIAX 600). RNA extraction was performed by using a Macherey-Nagel Nucleo Spin RNA kit (Düren, Germany) according to the manufacturer's guidelines. cDNA synthesis was then carried out by using a Bio-Rad iScript Reverse Transcription Supermix for RT-qPCR (Biorad, Hercules, CA, USA). A high throughput qPCR-based microfluid dynamic array technology (Fluidigm) was performed to evaluate the changes in gene expression. The converted cDNA was pre-amplified by using Applied Biosystems PreAmp Master Mix, first, one cycle 95 °C for 10 min, then 10 cycles, 95 °C for 15 s and finally one cycle, 60 °C for 4 min. The pre-amplified cDNA was diluted accordingly and was then used to perform the qPCR in Fluidigm dynamic array chip following the manufacturer's protocol with the aid of applicant specialists. The chip was run in the BioMark Instrument for qPCR at 95 °C for 10 min, followed by 40 cycles at 95 °C for 15 sec and lastly at 60 °C for 1 min. *Gapdh* was selected as a house-keeping gene as its expression were the most stable in kidneys. Taqman primers and probes for *Gapdh, Scnn1a* (*Enac-α*), *Scnn1b* (*Enac-β*) and *Scnn1g* (*Enac-γ*) were purchased from Applied Biosystems, CA, USA. All experiments were carried out in triplicates and the data were analyzed by using a Fluidigm Real-Time PCR analysis software.

## Quantification of ENaC protein levels in kidneys by Western blotting

The frozen kidneys, obtained from $n = 4$ rats per group were weighted, cut and the tissues submerged in RIPA buffer solution (BioVision, CA, USA) containing a protease inhibitor (BioVision, CA, USA), then disrupted using a rotor-stator homogenizer (Heidolph DIAX 600). The total proteins were obtained by centrifugation at 14,000× g for 15 min at 4 °C and protein concentration was quantified by a Thermo Scientific Micro BCA Protein Assay Kit (Rockford, IL, USA) following the manufacturer guidelines. Equal amount of proteins were added into a loading dye, boiled for 5 min and loaded onto a 10% SDS-PAGE gel. Following separations, the proteins were transferred onto a polyvinylidene fluoride

(PVDF) membrane (Bio-Rad, USA), then blocked in 5% bovine serum albumin (BSA) (Sigma-Aldrich, USA) for 60 min at room temperature. The membranes were then probed with primary antibodies against $\alpha$-ENaC, $\beta$-ENaC, $\gamma$-ENaC (sc-22239, sc-25354, sc-21014, Santa Cruz Biotechnology, CA, USA respectively) for 90 min at room temperature. This was followed by incubation in horseradish peroxidase (HRP) conjugated secondary antibody (Santa Cruz Biotechnology, Santa Cruz, CA, USA) for 60 min at room temperature. The blots were then developed by using a Thermo Scientific Super Signal West Pico Chemiluminescent Substrate (Rockford, USA) according to the manufacturer's manual. Chemiluminescent signals were captured by using a highly sensitive CCD camera-based imager (BioSpectrum Imaging System). The band intensity of each target was analyzed by using Image J software. GAPDH (Santa Cruz Biotechnology) was used as endogenous control. The experiment was performed in triplicate and the average ratios of target protein/GAPDH band intensity were then determined.

## Whole body perfusion fixation and detection of ENaC distribution in kidneys by immunofluorescence

Another set of animals ($n = 4$ rats per group), which were duplicate of those used for Western blotting and qPCR analyses, were anaesthetized with ketamine/xylazine anesthesia and a transcardial perfusion was performed with injection of 4% paraformaldehyde (PFA) in phosphate buffer saline (PBS). The animals were then killed and the kidneys were removed and fixed in 30% sucrose for three days. The tissues were then processed and embedded in a paraffin wax. The kidneys were sectioned coronally into 5 $\mu$m by using a microtome. Sections were then mounted onto poly-L-lysine coated glass slides. Sections were then de-parafinized and antigens were retrieved by boiling the sections in 1mM EDTA (BioVision InC.), pH 8.0 containing 0.05% (v/v) Tween20 (Sigma Aldrich). Sections were then blocked with appropriate 10% (v/v) normal serum (Santa Cruz Biotechnology) in PBS for 1 h at room temperature and were then incubated with target primary antibodies (as above) at the ratio of 1:100 at 4 °C, overnight. After three times of washing with PBS, sections were then incubated in appropriate fluorophore conjugated secondary antibodies at a dilution of 1:200, for 1 h at room temperature and then sealed with UltraCruz mounting medium (Santa Crus Biotechnology). Secondary antibodies used in this experiment were goat anti-mouse IgG FITC (sc-2010, Santa Crus Biotechnology, CA, USA), donkey anti-goat lgG (H+L) conjugated with DyLight 550 and donkey anti-rabbit lgG (H+L) conjugated with DyLight 488 (SA5-10087, SA5-10038, Thermo Scientific, Rockford, IL, USA). The images were viewed and captured by using a confocal laser scanning microscope (Leica TCS SP5 II) at a fixed exposure time. All experiments were carried out in four (4) replicates and the representative images were selected. All images were taken at 63x magnification and the slides without tissue were used as white balancer.

## Measurement of serum testosterone level by enzyme-linked immunoassay (ELISA)

Immediately following sacrifice and prior to transcardial perfusion, the blood from $n = 8$ rats per group was collected via heart puncture and the serum was separated by centrifugation at $3,500 \times$ g, 25 °C for 10 min. Levels of testosterone in serum were analysed

by enzyme-linked immunosorbent assay (ELISA) kit (BioSource International, Inc., Camarillo, CA, USA), according to the manufacturer guideline.

## Statistical analysis

All data were analyzed by SPSS software with mean $\pm$ standard error of mean (S.E.M) obtained. Statistical significant between groups was evaluated by Student $t$-test followed by one-way analysis of variance (ANOVA). Statistically significant difference were denoted as $p < 0.05$. Tukey's post-hoc test was used to examine adequacy of the samples and all values were >0.08 which indicate adequate sample size.

# RESULTS

## Testosterone increases $\alpha$-ENaC protein and mRNA levels in kidneys

In Fig. 1A, the levels of $\alpha$-*Enac* mRNA were highest following testosterone-only treatment. Co-treatment with flutamide or finasteride caused $\alpha$-*Enac* mRNA levels to decrease ($p < 0.05$). The effect of flutamide was greater in rats which received 125 μg/kg/day testosterone when compared to 250 μg/kg/day testosterone. However, the effect of finasteride in rats treated with 125 μg/kg/day testosterone- was not significantly different when compared to its effect in rats treated with 250 μg/kg/day testosterone.

In Figs. 1B and 1C, expression levels of $\alpha$-ENaC protein was highest in rats which received 250 μg/kg/day testosterone. In these rats, co-administration of flutamide or finasteride caused $\alpha$-ENaC protein expression level to decrease ($p < 0.05$). Flutamide effect was greater in rats treated with 125 μg/kg/day testosterone than in rat treated with 250 μg/kg/day testosterone. However, following co-administration of finasteride, no significant difference in $\alpha$-ENaC protein expression level was observed between rats which received 125 mg/kg/day testosterone and rats which received 250 μg/kg/day testosterone.

## Testosterone increases $\beta$-ENaC protein and mRNA levels in kidneys

In Fig. 2A, the levels of $\beta$-*Enac* mRNA were highest in rats which received testosterone-only treatment. In these rats, co-administration of flutamide or finasteride resulted in $\beta$-*Enac* mRNA levels to decrease ($p < 0.05$). The effect of flutamide was greater in rats which received 125 μg/kg/day testosterone when compared to rats which received 250 μg/kg/day testosterone. However, co-administration of finasteride did not cause $\beta$-*Enac* mRNA levels in rats which received 125 μg/kg/day testosterone to be difference from rats which received 250 μg/kg/day testosterone.

In Figs. 2B and 2C, the levels of expression of $\beta$-ENaC protein were highest in rats which received 250 μg/kg/day testosterone. In these rats, co-administration of flutamide or finasteride resulted in $\beta$-ENaC protein expression level to decrease ($p < 0.05$). The effects of flutamide were greater in rats which received 125 μg/kg/day testosterone when compared to rats which received 250 μg/kg/day testosterone. However, co-administration of finasteride did not cause $\beta$-ENaC protein expression level in rats which received 125 μg/kg/day testosterone to be different from rats which received 250 μg/kg/day testosterone.

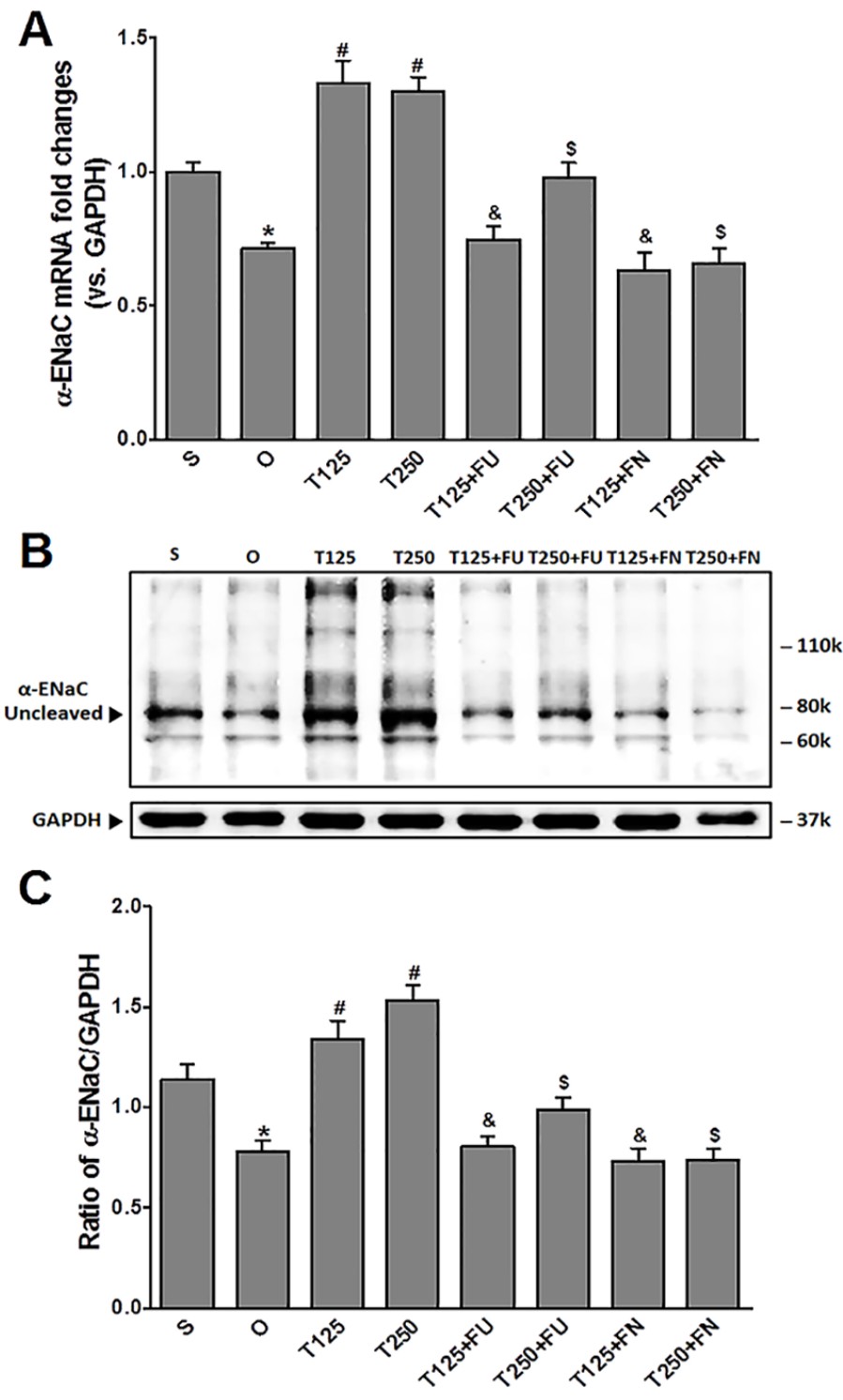

**Figure 1** **(A) $\alpha$-Enac mRNA level (B) whole membrane image of $\alpha$-ENaC protein band and (C) ratio of $\alpha$-ENaC/GAPDH protein band intensity in kidneys.** Values represent mean $\pm$ S.E.M of four rats. $^*p <$ 0.05 compared to sham-operated rats, $^\#p < 0.05$ compared to orcidectomized control rats, $^\&p < 0.05$ compared to T125. $^\$p < 0.05$ compared to T250. S, sham operated; O, orchiectomized non-treated; T125, 125 $\mu$g/kg/day testosterone-treated; T250, 250 $\mu$g/kg/day testosterone-treated rats; FU, flutamide; FN, finasteride. Molecular weight of $\alpha$-ENaC protein = 78 kDa (cleaved).

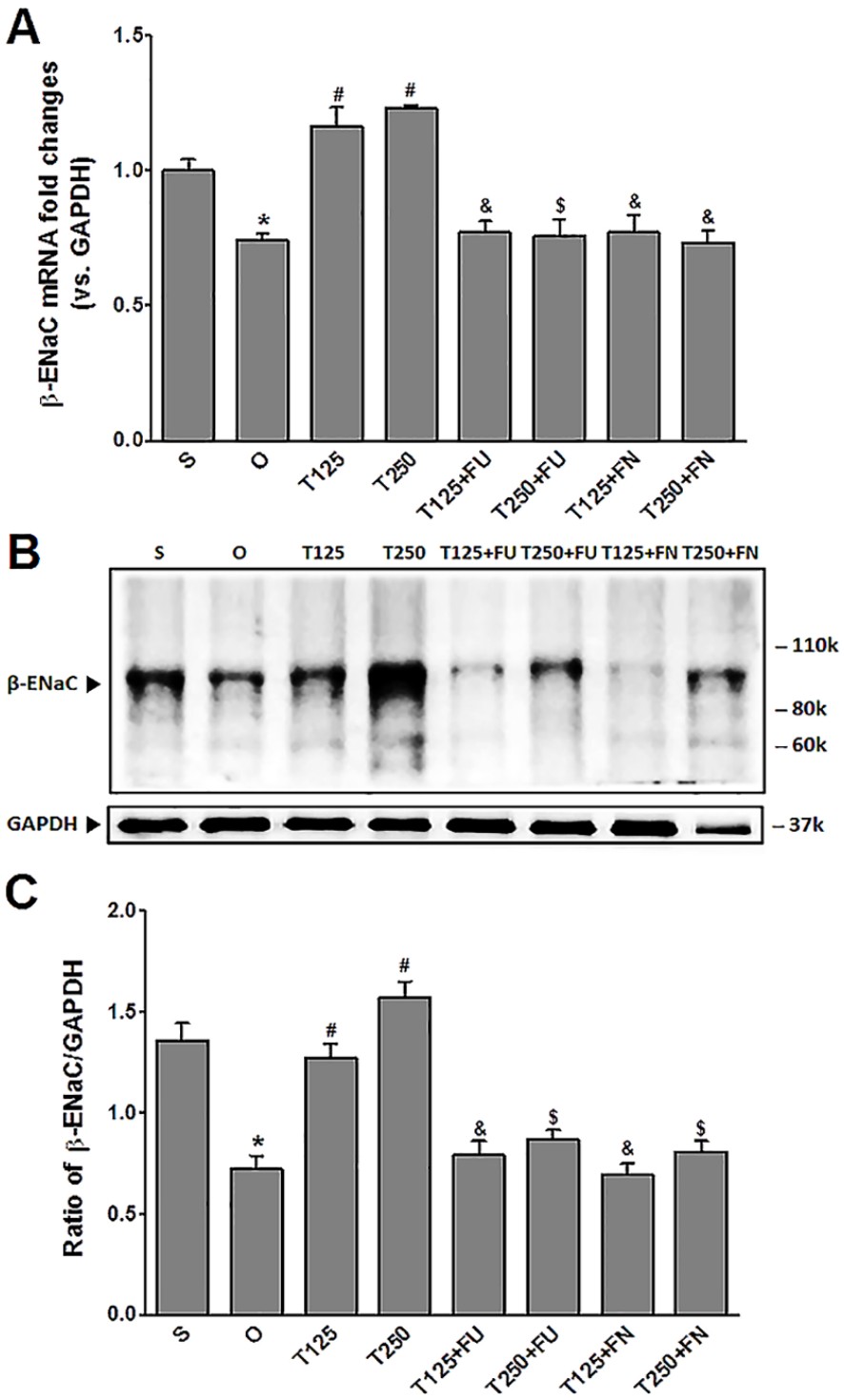

**Figure 2** **(A) β-Enac mRNA level (B) whole membrane image of β-ENaC protein band and (C) ratio of β-ENaC/GAPDH protein band intensity in kidneys.** Values represent mean ± S.E.M of four rats. $^*p < 0.05$ compared to sham-operated rats, $^\#p < 0.05$ compared to orcidectomized control rats, $^\&p < 0.05$ compared to T125. $^\$p < 0.05$ compared to T250. S, sham operated; O, orchiectomized non-treated, T125, 125 μg/kg/day testosterone-treated, T250, 250 μg/kg/day testosterone-treated rats; FU, flutamide; FN, finasteride. Molecular weight of β-ENaC protein = 99 kDa.

### Testosterone increases $\gamma$-ENaC protein and mRNA levels in kidneys

In Fig. 3A, the levels of $\gamma$-Enac mRNA were highest in rats which received 250 µg/kg/day testosterone. In these rats, co-administration of flutamide or finasteride resulted in $\gamma$-Enac mRNA levels to decrease ($p < 0.05$). The effect of flutamide was not significantly different between rats which received 125 µg/kg/day testosterone and rats which received 250 µg/kg/day testosterone. Similarly, the effect of finasteride in rats which received 125 µg/kg/day testosterone was not significantly different from its effect in rats which received 250 mg/kg/day testosterone.

In Figs. 3B and 3C, the levels of $\gamma$-ENaC protein was highest in rats which received 250 µg/kg/day testosterone. In these rats, co-administration of flutamide or finasteride resulted in $\gamma$-ENaC protein expression level to decrease ($p < 0.05$). Co-administration of flutamide with testosterone did not cause $\gamma$-ENaC protein expression level in rats which received 125 µg/kg/day testosterone to be different from rats which received 250 µg/kg/day testosterone. In contrast, co-administration of finasteride with testosterone caused greater decrease in the level of this protein in rats which received 250 µg/kg/day testosterone when compared to rats which received 125 µg/kg/day testosterone.

### Distribution of $\alpha$-ENaC protein in nephrons

In Fig. 4, $\alpha$-ENaC protein was seen to be highly distributed in distal tubules and collecting ducts of rats which received 125 and 250 µg/kg/day testosterone. In these rats, co-administration of flutamide or finasteride resulted in $\alpha$-ENaC protein distribution level to be relatively lower (Fig. 5).

### Distribution of $\beta$-ENaC protein in nephrons

In Fig. 4, $\beta$-ENaC protein was highly distributed in distal tubules and collecting ducts in rats which received 250 µg/kg/day testosterone. However, co-administration of flutamide or finasteride with testosterone resulted in relatively lower $\beta$-ENaC protein distribution level when compared to rats which received testosterone-only treatment (Fig. 6).

### Distribution of $\gamma$-ENaC protein in nephrons

In Fig. 4, $\gamma$-ENaC protein was highly distributed in distal tubule and collecting ducts in rats which received 125 and 250 µg/kg/day testosterone. However, co-administration of flutamide with testosterone resulted in relatively lower $\gamma$-ENaC protein distribution when compared to testosterone-only treatment (Fig. 7). Similar effects could be seen in rats following concomitant co-administration of finasteride with testosterone.

### Serum testosterone level

In Table 1, administration of 125 µg/kg/day testosterone resulted in serum level of this hormone to be 13.5 fold greater than its level in non-testosterone-treated orchiectomised rats. Treatment with 250 µg/kg/day testosterone resulted in serum testosterone level to be 16.5 fold greater than in non-testosterone treated orchiectomized rats.

## DISCUSSION

To the best of our knowledge, this study has for the first time revealed that testosterone was able to up-regulate expression of ENaC subunits' ($\alpha$, $\beta$ and $\gamma$) proteins and mRNAs in the

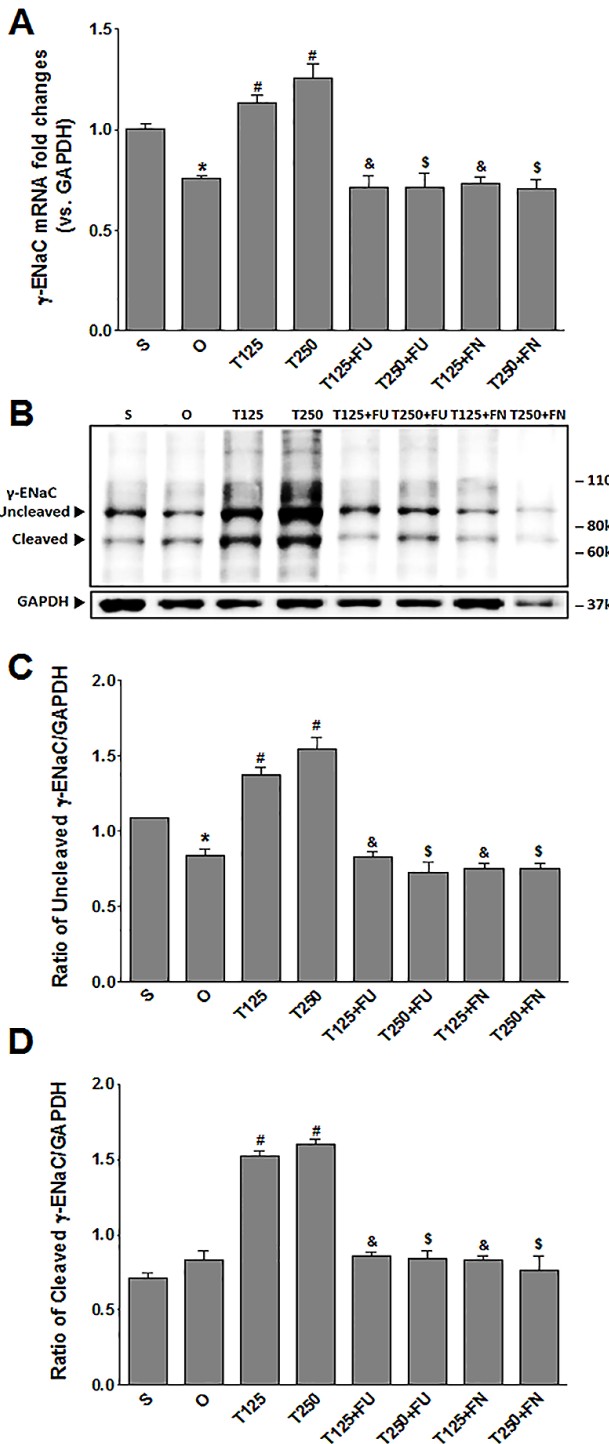

**Figure 3** **(A) γ-Enac mRNA level (B) whole membrane image of γ-ENaC protein band and (C) ratio of γ-ENaC/GAPDH protein band intensity in kidneys.** γ-ENaC possesses a cleaved and an un-cleaved forms. Values represent mean $\pm$ S.E.M of four rats. $*p < 0.05$ compared to sham-operated rats, $^{\#}p < 0.05$ compared to orcidectomized control rats, $^{\&}p < 0.05$ compared to T125. $^{\$}p < 0.05$ compared to T250. S, sham operated; O, orchiectomized non-treated; T125, 125 μg/kg/day testosterone-treated; T250, 250 μg/kg/day testosterone-treated rats; FU, flutamide, FN, finasteride. Molecular weight of γ-ENaC = 75 kDa (cleaved/unglycosylated).

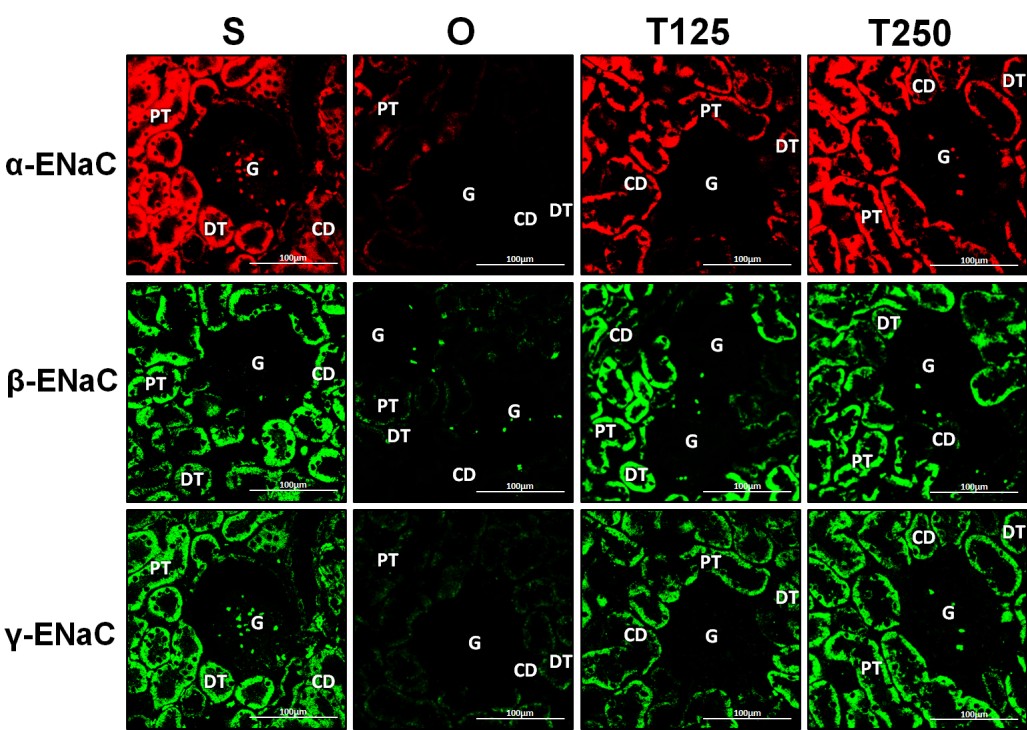

**Figure 4** **Immunofluorescenceimages showing distribution of α, β and γ-ENaC proteins in nephrons.** Red or green fluorescence signals indicate the sites where ENaC subunit proteins were expressed. G, glomeruli; PT, prximal tubule; DT, distal tubule, CD, collecting duct. S, sham operated; O, orchiectomized non-treated; T125, 125 µg/kg/day testosterone-treated; T250, 250 mg/kg/day testosterone-treated rats. Scale bar = 100 µM.

**Table 1** **Serum level of testosterone in different experimental groups.** Values were expressed as mean ± S.E.M of eight (8) different observations.

| Groups | Serum testosterone levels (ng/ml) |
|---|---|
| Sham | $2.89 \pm 0.38$ |
| ORX | $0.25^{a} \pm 0.20$ |
| T125 | $3.37^{b} \pm 0.40$ |
| T250 | $4.12^{b} \pm 0.55$ |

**Notes.**

[a] $p < 0.05$ compared to shamoperated rats.

[b] $p < 0.05$ compared to orchidectomized control rats.

Sham, Sham-operated; ORX, Orchidectomized control; T125, 125 µg/kg/day testosterone; T250, 250 µg/kg/day testosterone.

kidneys of orchidectomised adult male rats *in-vivo*. The dose-dependent increase in serum testosterone levels was observed when testosterone was subcutaneously injected to these rats at increasing doses which suggested that this hormone most likely was not metabolized as it bypassed the first-pass effect. It was found that equal amount of $\alpha$-ENaC protein was expressed in the orchidectomised male rats' kidneys following administration of 125 and 250 µg/kg/day testosterone, however higher expression of $\beta$ and $\gamma$ ENaC proteins were found in the kidneys of rats which received 250 µg/kg/day testosterone compared

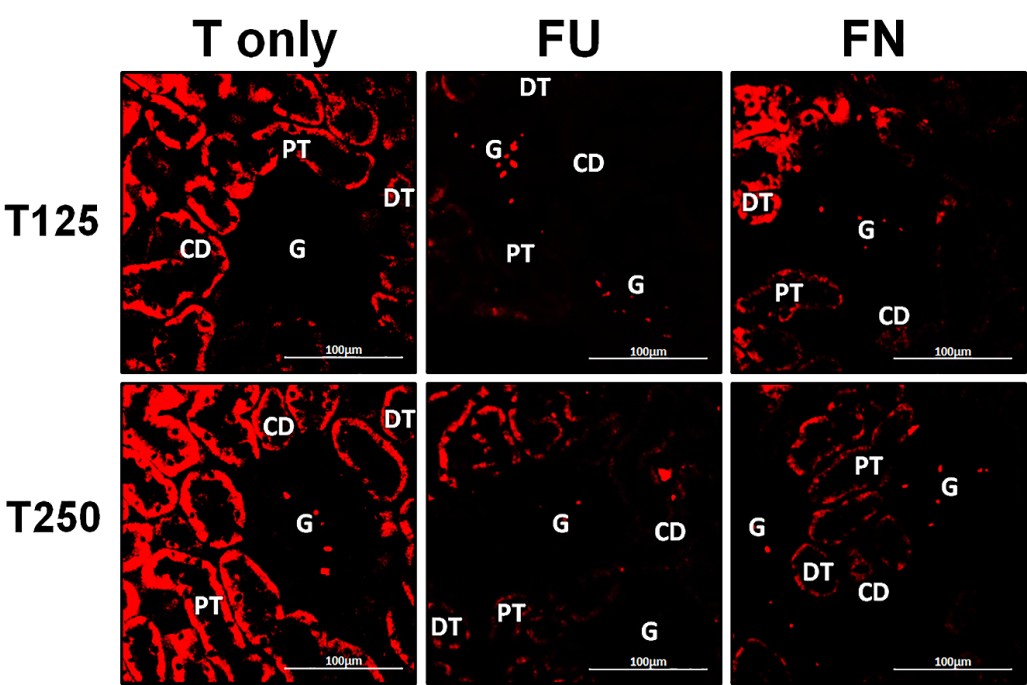

**Figure 5** **Effects of flutamide and finasteride on expression of α-ENaC in nephrons.** Red fluorescence signals indicate sites where ENaC subunit proteins were expressed. G, glomeruli; PT, prximal tubule; DT, distal tubule; CD, collecting duct. S, sham operated; O, orchiectomized non-treated; T125, 125 µg/kg/day testosterone-treated; T250, 250 µg/kg/day testosterone-treated rats. Scale bar = 100 µM.

to 125 µg/kg/day testosterone. The latter findings were consistent with the findings by *Kienitz et al.* (*2006*) who reported that administration of testosterone at 500 mg/kg/ in male rats resulted in high level of α-ENaC protein in kidneys. Another study has revealed that the levels of *α-ENaC* mRNA in human kidney cell line was enhanced by testosterone, however this hormone was found to have no significant effect on $\beta$ and *γ-ENaC* mRNA levels (*Quinkler et al.*, *2005*). In contrast, we found that the levels of $\beta$ and $\gamma$- ENaC proteins and mRNAs were highly up-regulated in the orchidectomised male rats' kidneys by testosterone. We postulated that co-expression of all ENaC subunits ($\alpha$, $\beta$ and $\gamma$) would result in a fully operating channel as their co-existence was required for the maximal ENaC channel function (*Hamm, Feng & Hering-Smith*, *2010*).

In this study, we have found that the level of expression of $\alpha$, $\beta$ and $\gamma$-ENaC proteins and mRNAs in the kidney was significantly decreased following co-administration of flutamide with testosterone. The effect of flutamide on $\alpha$ and $\beta$-ENaC but not $\gamma$-ENaC was found greater in rats which received 125 µg/kg/day when compared to 250 µg/kg/day testosterone. The reason behind this effect was unknown; however, we postulated that following high dose testosterone treatment i.e., at 250 µg/kg/day, some of this hormone might be aromatized to estradiol in the adipose tissue, penis and brain (*Schulster, Bernie & Ramasamy*, *2016*) while some bind to the tissue androgen receptor. The aromatized form of testosterone i.e., estradiol could cause increased in α-ENaC levels in the kidney (*Gambling et al.*, *2004*). This could perhaps explain the reason why lesser inhibition was

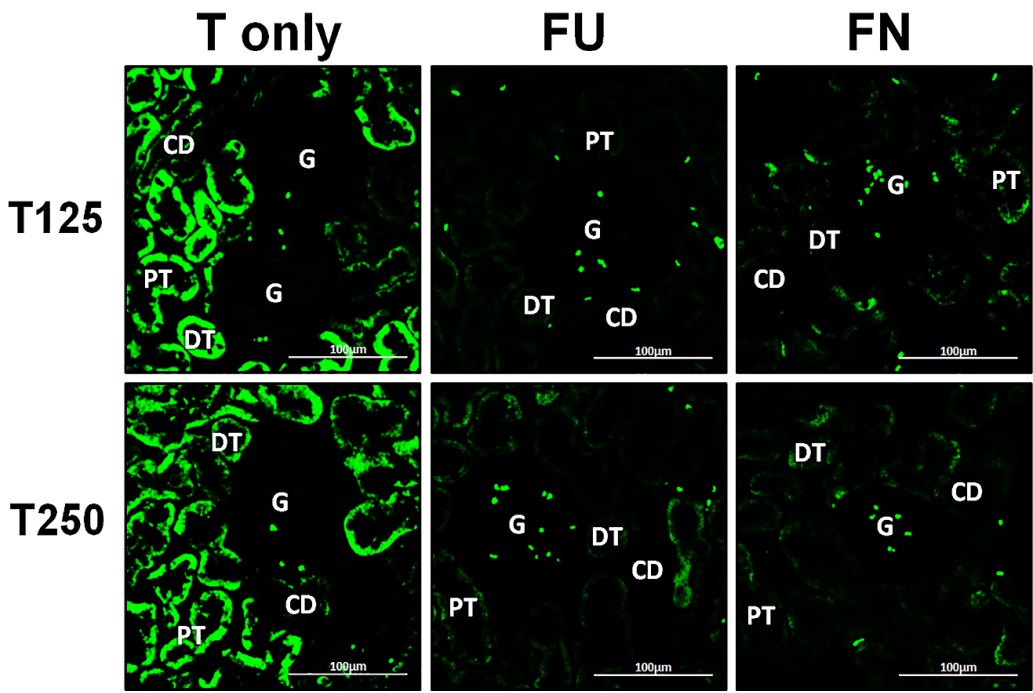

**T only**      **FU**      **FN**

**Figure 6 Effects of flutamide and finasteride on expression of $\beta$-ENaC in nephrons.** Green fluorescence signals indicate sites where ENaC subunit proteins were expressed. G, glomeruli; PT, prximal tubule; DT, distal tubule; CD, collecting duct. S, sham operated; O, orchiectomized non-treated; T125, 125 $\mu$g/kg/day testosterone-treated; T250, 250 $\mu$g/kg/day testosterone-treated rats. Scale bar = 100 $\mu$M.

seen following high dose testosterone treatment. In the meantime, at high dose, there was a possibility that testosterone could exert a non-genomic effect which was not inhibited by flutamide (*Mokhtar et al.*, *2014*). Finasteride was found to inhibit testosterone effect in causing increase in ENaC subunits expression in the kidneys, which suggested that DHT was involved. In the absence of DHT, levels of ENaC subunits in kidneys would be markedly decreased. Besides DHT, other hormone which was also known to cause expression level of $\alpha$, $\beta$ and $\gamma$-ENaC subunits in the kidneys to increase is aldosterone, which was reported to induce redistribution of all ENaC subunits to the apical membrane of kidneys' distal tubule and collecting ducts (*Stachowiak, Nussdorfer & Malendowicz*, *1991*; *Masilamani et al.*, *1999*). Therefore, redistribution of all ENaC subunits to the apical membrane of distal tubule and collecting duct under testosterone influence might produce similar effect to that reported under aldosterone, in which this would result in fully functioning channels (*Masilamani et al.*, *1999*). The resultant effect would lead to increase in Na$^+$ reabsorption. We have found that ENaC subunits' protein were also found to be expressed in the cytoplasm under testosterone influence, in which these distributions could represent the channels at various stages of processing.

The effect of testosterone on ENaC subunit expression as observed in our study could help to explain the mechanisms underlying this hormone effects on kidneys' Na$^+$ handling as have been previously reported (*Reckelhoff, Zhang & Granger*, *1998*). Based on what

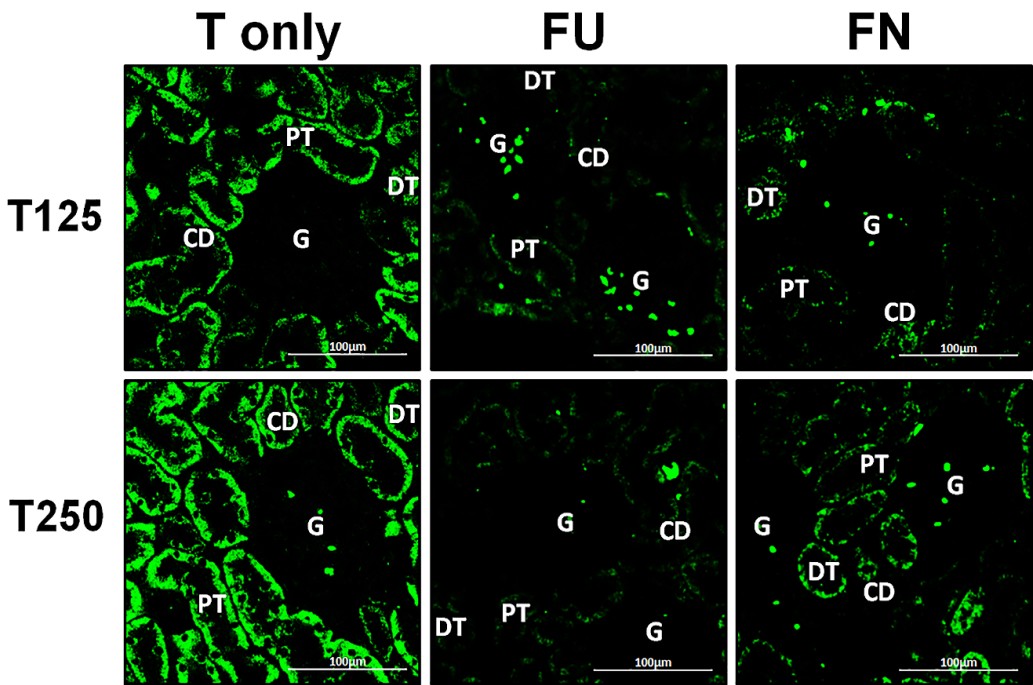

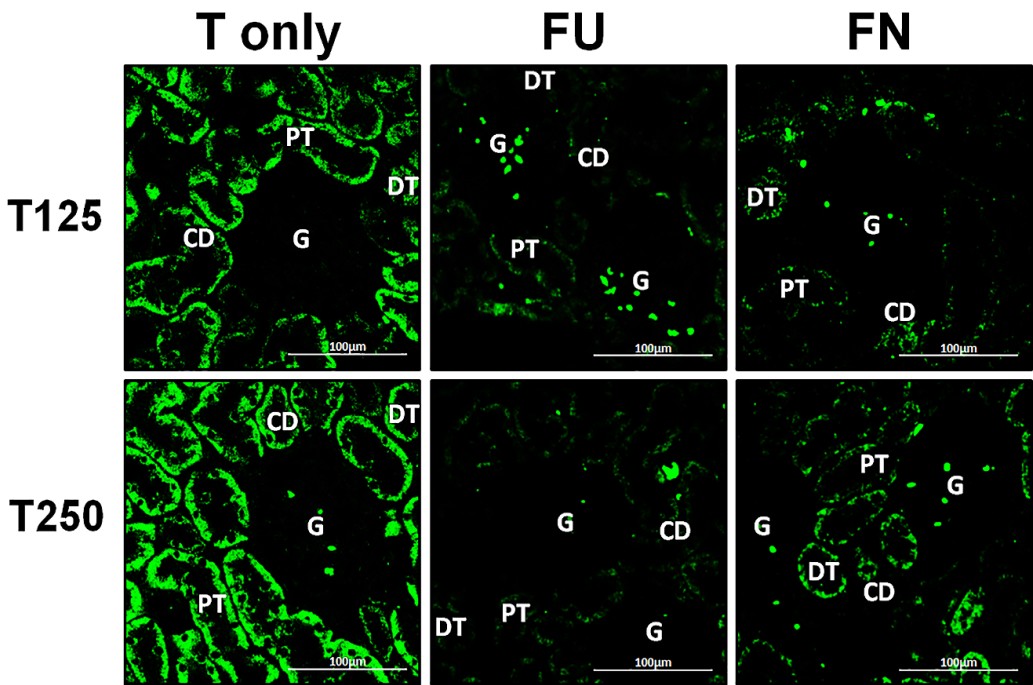

**Figure 7 Effects of flutamide and finasteride on expression of $\gamma$-ENaC in nephrons.** Green fluorescence signals indicate sites where ENaC subunit proteins were expressed. G, glomeruli; PT, prximal tubule; DT, distal tubule; CD, collecting duct. S, sham operated; O, orchectomized non-treated; T125, 125 $\mu$g/kg/day testosterone-treated; T250, 250 $\mu$g/kg/day testosterone-treated rats. Scale bar = 100 $\mu$M.

has been known, we have provided additional information in which in kidneys of male rat model, testosterone was also found to enhance $\beta$ and $\gamma$-ENaC subunit expression in addition to the already known increase in $\alpha$-ENaC expression (*Quinkler et al.*, *2005*). These testosterone effects would likely result in a fully functioning channel. However, more works need to be done in order to confirm the optimal ENaC channel function at the apical membrane of the distal tubule and collecting duct epithelia for example by using a patch clamp in which kidney epithelial cells obtained from animals treated with testosterone will be exposed to a specific ENaC inhibitor such as amiloride (*Edinger et al.*, *2012*). If in the case where ENaC channels are functioning, inhibition by amiloride will result in reduced Na$^+$ conductance. Additionally, the involvement of DHT can also be confirmed by administering this compound directly to the rats and the involvement of genomic pathway in mediating testosterone effect on ENaC expression can be confirmed by using androgen receptor knock-out animal model.

Our results might be clinically relevant. The role of testosterone in causing development of hypertension has long been debated. So far there has been no studies implicating direct involvement of this hormone in the development of this disease in males. Furthermore, molecular mechanisms underlying hypertension development that were linked to testosterone remain largely unknown. Several evidences showed that testosterone has direct link to blood pressure. *Reckelhoff, Zhang & Granger* (*1998*) reported that in intact spontaneous male hypertensive (SHR) rats, elevated blood pressure was linked

to high serum testosterone levels and this was further confirmed from the observations that ovariectomized female SHR rats that was given testosterone had increase in blood pressure. Furthermore, studies have shown that natriuresis were markedly reduced in intact male and testosterone-treated ovariectomised female SHR rats which indicated that higher amount of $Na^+$ was reabsorbed in the kidney under the influence of this hormone (*Reckelhoff, Zhang & Granger*, *1998*). Similar findings were reported in humans whereby serum testosterone levels correlate with the blood pressure (*Khaw & Barrett-Connor*, *1988*).

In conclusions, our findings have provided evidences which might support the role of testosterone in causing a relatively higher blood pressure in males than females. Increased in kidney $Na^+$ reabsorption which might occur secondary to testosterone-induced up-regulation of ENaC might predispose the males to hypertension and could perhaps be the reason why males have higher incidence of this disease compared to age-matched females before menopause. Finally, the differential effect of testosterone on $\alpha$, $\beta$ and $\gamma$-EnaC subunits expression in the kidney might contribute towards gender differences in blood pressure regulation (*Reckelhoff*, *2001*; *Reckelhoff et al.*, *1999*).

### Funding
This work was funded by the University of Malaya UMRG research fund (RPO11/13HTM) and a PPP grant (PG002-2014B). The funders had no role in study design, data collection and analysis, decision to publish, or preparation of the manuscript.

### Grant Disclosures
The following grant information was disclosed by the authors:
University of Malaya UMRG research fund: RPO11/13HTM.
PPP grant: PG002-2014B.

### Competing Interests
The authors declare there are no competing interests.

### Author Contributions
- Su Yi Loh performed the experiments, analyzed the data, prepared figures and/or tables.
- Nelli Giribabu performed the experiments, prepared figures and/or tables.
- Naguib Salleh conceived and designed the experiments, analyzed the data, contributed reagents/materials/analysis tools, wrote the paper, reviewed drafts of the paper.

### Animal Ethics
The following information was supplied relating to ethical approvals (i.e., approving body and any reference numbers):

Institutional Animal Care and Use Committee (IACUC), University of Malaya (ethics number: 2014-05-07/physio/R/NS).

## Data Availability

The raw data has been supplied as Supplemental Datasets.

## Supplemental Information

Supplemental information for this article can be found online at http://dx.doi.org/10.7717/peerj.2145#supplemental-information.

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
