# Peer review of "Sub-chronic testosterone treatment increases the levels of epithelial sodium channel (ENaC)-α, β and γ in the kidney of orchidectomized adult male Sprague–Dawley rats"

_PeerJ, doi:10.7717/peerj.2145_

## Round 0.1 · original submission · Minor Revisions

In addition to the reviewers’ comments please clarify the following points:

1) The Methods state that group sizes were n=8. Please specify if all rats were used in one experiment, or if the 8 rats per group were generated over more than one experiment.

2) Perfusion for histological analysis precludes the use of the rat for any other assays. Given that six rats were used for Table 1, can we assume that two rats were used for Immunohistochemistry? Are the immunohistochemistry data shown representative of an n=2?

3) Note that expression does not equate function, as implied in the third-last paragraph of the Discussion, line 2 “..which indicated that they are involved in Na+ reabsorption...”. Modify this sentence into a more speculative statement.

Concerning the requests from Revier#2, please note that additional experiments will not be necessary. However, please provide a full description of the data shown in Reference #7 and Kienitz et al, 2009. In addition, for all points raised by Reviewer#2, modify the text to adjust your conclusions, be explicit about the limited conclusions that can be drawn from the experiments shown in this manuscript with respect to e.g. the role of androgen receptor and gender specificity. (The Discussion may include a section on the future experiments that need to be conducted to address these open questions.) Furthermore, the Conclusion sentence in the Abstract should be split into a conclusion sentence limited on the actual data, which can be followed by a speculation on the potential effect on Na+ reabsorption and blood pressure.

Reviewer 1 ·

Basic reporting

The manuscript was well written. The introduction and background were sufficient to understand how the research question fitted into the hypertension field, and the knowledge gap was identified. Figures are relevant and well described.

Experimental design

The research question was well defined and relevant, The methods were detailed and can be easily replicated by other labs, which is not commonly seen in the published paper now a days as Vice President emphasized the importance of the data reproducibility recently related to Moonshot initiative.The experiments were well designed (e.g. included appropriate controls and experimental groups).

Validity of the findings

Conclusion was well stated, and supported by the results. While mRNA and protein levels are quantitative, immunostaining data strongly support the conclusion.

Figure 2C and Figure 3D were not normalized to S group as 1. It should be presented as that in Figure 1C.

Additional comments

No comments

Reviewer 2 ·

Basic reporting

Addressed below in general comments for the author.

Experimental design

No comments.

Validity of the findings

Addressed below in general comments for the author.

Additional comments

The current study by Yi et al investigates the effects of testosterone on expression of α, β, and γ epithelial sodium channel (ENac) proteins and mRNAs in the kidneys of male Sprague Dawley rats. The results attempt to demonstrate that testosterone induced increases in these ENaC subunits, which could enhance sodium reabsorption leading to an increase in blood pressure. This study is missing some key references/studies to support its conclusion that their findings could help explain mechanisms underlying higher blood pressure in males compared to females.

Specific comments:
1. Page 6 (line 63-64): The authors state “Limited findings revealed that expression of α-ENac mRNA in kidney cell line could be affected by testosterone (Reference 7 – Quinkler et al).” However, reference 7 also showed that testosterone increased renal α-ENac mRNA expression in male rats. The authors should include this line of evidence here.

2. Page 6 (line 72): The authors state “Additionally, the involvement of androgen receptor and DHT in mediating testosterone effects on these molecular parameters were also investigated.” The authors used flutamide and finasteride to demonstrate the inhibitory effect on testosterone-induced expression of ENaC subunits. However, more direct evidence would be to examine the expression level of the androgen receptor in these samples. A subset of the animals should also be treated with DHT to determine if a similar pattern of expression of the ENaC subunits were observed as compared with testosterone treatment.

3. Page 13 (section 3.7): Serum testosterone levels were measured; however, the relevance of this finding was not addressed in the Discussion section.

4. Page 15 (line 302): The authors state “In conclusions, our findings could help to explain mechanisms underlying higher blood pressure in males compared to females.” To come to such a conclusion, the authors should have evaluated the effects of sex steroids on ENaC expression in female rats as well. This is especially since previous studies by Kienitz et al (Horm Metab Res 2009; 41(5): 356-362) have demonstrated sex-specific regulation of ENaC and androgen receptor in the female rat kidney, which is an extension to their studies on the male rat kidney (reference 7). Because the current study demonstrated data that is in contrast to the Quinkler et al study with respect to the effects of testosterone on β- and γ-ENaC expression, it would be equally important to determine these effects in the female rat kidneys to compare their findings with Kienitz et al.

---

## Round 0.2 · accepted · Accept

The editorial comments have been adequately addressed.

Reviewer 2 ·

Basic reporting

No comments

Experimental design

No comments

Validity of the findings

No comments

Additional comments

The authors have adequately addressed the reviewers' comments and the current revised version of this manuscript is acceptable.